# putEMG—A Surface Electromyography Hand Gesture Recognition Dataset

**DOI:** 10.3390/s19163548

**Published:** 2019-08-14

**Authors:** Piotr Kaczmarek, Tomasz Mańkowski, Jakub Tomczyński

**Affiliations:** Institute of Control, Robotics and Information Engineering - Poznan University of Technology, Piotrowo 3A, 60-965 Poznań, Poland

**Keywords:** sEMG, dataset, gesture recognition, hand, human-machine interface, wearable

## Abstract

In this paper, we present a putEMG dataset intended for the evaluation of hand gesture recognition methods based on sEMG signal. The dataset was acquired for 44 able-bodied subjects and include 8 gestures (3 full hand gestures, 4 pinches and idle). It consists of uninterrupted recordings of 24 sEMG channels from the subject’s forearm, RGB video stream and depth camera images used for hand motion tracking. Moreover, exemplary processing scripts are also published. The putEMG dataset is available under a Creative Commons Attribution-NonCommercial 4.0 International (CC BY-NC 4.0). The dataset was validated regarding sEMG amplitudes and gesture recognition performance. The classification was performed using state-of-the-art classifiers and feature sets. An accuracy of 90% was achieved for SVM classifier utilising RMS feature and for LDA classifier using Hudgin’s and Du’s feature sets. Analysis of performance for particular gestures showed that LDA/Du combination has significantly higher accuracy for full hand gestures, while SVM/RMS performs better for pinch gestures. The presented dataset can be used as a benchmark for various classification methods, the evaluation of electrode localisation concepts, or the development of classification methods invariant to user-specific features or electrode displacement.

## 1. Introduction

Electromyography (EMG) is a well-established method of muscle activity analysis and diagnosis. A go-to approach with creating a user-friendly human machine-interfaces (HMIs) would be utilising the surface electromyography (sEMG), where non-invasive, on-skin electrodes are used to register muscle activity. Despite numerous attempts, its application in HMIs is very limited. Currently, excluding neuroprostheses, there are no commercial applications utilising interfaces based on the EMG signal. So far, the most popular attempt to commercialise sEMG-driven interface is the no longer manufactured Myo armband by Thalmic Labs, enabling recognition of few hand gestures. Several problems related to the development of interfaces of this type exist. The first crucial issue is the development of electrodes ensuring constant input impedance while providing sufficient user comfort and ease of use in conditions outside of a laboratory during prolonged usage [1,2,3]. Secondly, a commercial application has to meet a satisfactory level of gesture recognition accuracy, even when used by a large variety of subjects. The end-user product has to provide a low entry threshold—a long and complicated calibration procedure will discourage the user from an EMG-based HMI. However, these requirements can be difficult to achieve, as the EMG signal is strongly individual [4] and non-stationary [5]. Moreover, sEMG is susceptible to external factors such as additional mechanical loads [6,7,8]. Consequent re-wearing of the sEMG gesture recognition system will result in changes of the signal characteristics, as the precision of electrode placement and alignment is limited for an end-user device [9,10]. In laboratory conditions, the accuracy of an HMI can be greatly improved by training the classifier for each user and each device use. Several attempts are being made to solve these problems, nonetheless, the results are not satisfactory enough to be implemented in an end-user device [5,8,10,11,12]. Potentially, modern machine learning solutions used in the analysis of images and in big data solutions, like deep neural-networks (DNNs), can be exploited in tasks of creating effective sEMG-based hand gesture recognition device. However, this requires availability of ready to use large data collections and processing methods. Publicly available datasets, to a large extent, contribute to accelerating this process by enabling research teams to develop and compare methods in a reliable way [13].

Several datasets containing multi-channel sEMG recordings of hand gesture executions are publicly available. Table 1 presents a summary of selected datasets with free-of-charge access. The most extensive database is the NinaPro (Non-Invasive Adaptive Hand Prosthetics) containing seven separate sets of data [14,15,16,17]. NinaPro DB2 [16] contains data recorded for 40 people and 52 gestures respectively, with movements tracked with CyberGlove II. Due to a large number of gestures and participants, NinaPro DB2 collection enables the detection of discrete gestures, as well as the development of methods for continuous joint angle estimation [18].

Most of the published sEMG datasets lack in the number of gesture repetitions, which is vital for the development of robust recognition algorithms. Moreover, when performing each gesture several times in a row (denoted in Table 1 as repetitive) it is likely for the subject to perform the gesture in a very similar manner. As a consequence, due to a strong correlation between sEMGs signals, exceptional classification accuracy might be achieved but significant classifier overfitting can blur the conclusions. For several datasets, gestures were executed sequentially or in random order (denoted in Table 1 as sequential and random). While both these approaches reduce the risk of a subject performing each gesture identically, it is still vital to include a large number of repetitions. Due to a number of variations that can occur in a specific gesture, a repetition count as low as 10 can be insufficient, even if k-fold validation is used. The IEE EMG database [19] offers the largest number of gesture repetitions—each active gesture was repeated 32 times; however, data was acquired for only four subjects. The majority of sEMG datasets contain trials recorded only during a single session. The development of an EMG-based gesture recognition system requires taking into account both EMG signal individuality and long-term variations resulting from fatigue and physiological changes. Moreover, for the end-user device, subsequent re-wearing will cause electrode misalignment. Thus, datasets containing multiple sessions will have a positive impact on such system designs. Four databases listed in Table 1 contain data recorded during more than one session [10,20,21]. NinaPro DB6 [10] and Megane Pro [20] include data acquired for 10 people (7 and 15 gestures respectively), where the experiment was performed with 5-day intervals, twice each time.

The datasets listed in Table 1 represent two main measurement methods: High-Density sEMG 128–196 electrode arrays [21,24] and 10–16 electrode approach, where electrodes are often placed in anatomical points [15,16,17,19,20,22]. High-Density matrix setups provide exceptional insight into EMG signal formation processes and the nature of the phenomenon; however, they are not applicable when creating portable and wearable end-user gesture recognition systems. Databases containing signals recorded from lower electrode counts, placed in sparser points, are more likely to be used in process of development of such devices, as they provide signals from sensor configuration closer to a feasible end-user device. Real-time, synchronised gesture tracking is a crucial element of any usable dataset. Lack of gesture tracking during the experiment makes verification of performed gesture and proper labelling of the data difficult. The majority of datasets utilise CyberGlove II; however, glove stiffness introduces an additional mechanical load to the hand, which can lead to changes in EMG signal and create a discrepancy in classifier performance, when it is applied for gesture classification performed without the glove. Despite using CyberGlove II, in most databases, data re-labelling is based on the EMG signal itself—gesture duration is detected based on the onset or decay of the EMG signal activity [15,19]. No additional gesture tracking was used in NinaPro DB4 [17], NinaPro  DB7 [22], IEE EMG [19], and CapgMyo [21] datasets.

### Contribution

In this work, we would like to present the putEMG dataset (see Table 1 for a summary) containing sEMG data acquired using three 8-electrode (total of 24 electrodes) elastic bands placed around the subjects’ forearm. 7 active gestures, plus an *idle* state are included in the dataset. An important advantage, compared to existing datasets, is a large number of repetitions (20) of each gesture. Data was collected for 44 participants in identical experimental procedure, repeated twice (with one-week interval)—raising the number of each gesture repetition to 40. All gestures were performed by subjects in sequential or repetitive manners, with gesture duration of 1 s and 3 s. The presented dataset can be used to develop and test user-invariant classification methods, determine features resistance to both long-term changes in the EMG signal and inaccurate electrode re-fitting. Usage of sparse 24 matrix-like electrode configuration enables development of methods independent from electrode placement. We also propose a novel approach to ground-truth of gesture execution, as in the case of putEMG dataset hand movement was tracked using a depth sensor and an HD camera. Then, video stream was used for gesture re-labelling and the detection of possible subjects’ mistakes. In comparison to glove-like systems, this solution does not limit the freedom of hand movement and does not apply any additional mechanical load to the examined hand. The putEMG dataset is available at https://www.biolab.put.poznan.pl/putemg-dataset/. Exemplary Python usage scripts allowing for fast dataset adoption are provided as GitHub repositories, for more details see Appendix A.

## 2. putEMG Data Collection

### 2.1. Experimental Setup

A dedicated experimental setup was developed for the purposes of putEMG dataset acquisition. The setup was designed to allow for forearm muscle activity recording of a single subject with a wide range of anatomical features, especially arm circumference. Great emphasis was put on ease of use and electrostatic discharge protection as a large number of subjects participating in the experiment was planned. The putEMG data collection bench is visible in Figure 1.

For sEMG signal recording, a universal, desktop multi-channel biosignal amplifier, MEBA by OT Bioelettronica, was used. Data was sampled at 5120 Hz, with 12-bit analog to digital converter (ADC) resolution and gain of 200. Additionally, built-in analogue band-pass filter, with a bandwidth of 3 to 900 Hz, was applied in order to eliminate bias and prevent aliasing. Signals were recorded in monopolar mode with DRL-IN and Patient-REF electrodes placed in close proximity of the wrist of the examined arm.

Twenty-four electrodes were used while recording sEMG signals. Electrodes were fixed around subject right forearm using 3 elastic bands, resulting in a 3×8 matrix. Bands were responsible for uniformly distributing electrodes around the participant’s forearm, with 45 spacing. The first band was placed in approximately 14bevelledtrue of forearm length measuring from the elbow, each following band was separated by approximately 15bevelledtrue of forearm length. The first electrode of each band was placed over the ulna bone and numbered clockwise respectively. Following channel numbering pattern was used—elbow band {1–8}, middle band {9–16}, wrist band {17–24}. Schematics of armbands placement is presented in Figure 2a. Different band sets were used in order to compensate for differences in participants’ forearm diameter. The authors’ reusable wet electrode design was used. A 3D-printed electrode allows for simple fixing to the elastic band using snap joints. Each electrode contains a 10 mm Ag/AgCl-coated element and a sponge insert saturated with electrolyte. An exploded view of the electrode design is visible in Figure 2b.

Additionally, the setup included an HD Web Camera (Logitech C922) and a short-range depth sensor (RealSense SR300) with a close view of the subject’s hand. Both video feed and depth images are provided alongside the sEMG data. These devices allow for the detection of subject’s mistakes, the compensation of reaction time and the verification of the experiment course.

### 2.2. Procedure

putEMG dataset consists of a series of trials that include execution of 8 hand gestures (see Figure 3). The gesture choice was based on former studies, active poses showing the highest discrimination [9,25,26] and best fitting for convenient sEMG-based HMI design were selected. The set consists of 7 active gestures — *fist*, *flexion*, *extension*, and 4 pinch gestures where the thumb meets one of the remaining fingers (*pinch index*, *pinch middle*, *pinch ring*, *pinch small*—Figure 3e–h). The 8th gesture in the putEMG dataset is *idle*, during which subjects were asked not to move their hand, keep it stabilised and relax the muscles. A 3-s *idle* period always separates execution of active gestures. Each time an active gesture was held for 1 or 3 s, depending on trajectory configuration. Subjects were instructed to execute all gestures with their elbow resting on the armrest with forearm elevated at a 10–20 angle.

Each experiment procedure consisted of 3 trajectories. Trajectories were divided into action blocks between which the subject could relax and move hand freely, with no restrictions. This part was introduced in order to reduce possibility of cramps and give subject a chance to rest. Each *relax* period lasted for 10 s and is marked with a ( −1) label in putEMG data files. Each experiment includes the following trajectories:
**repeats_long** - 7 action blocks, each block contains 8 repetitions of each active gesture:*[relax] 0-1-0-1-0-1-0-1-0-1-0-1-0-1-0-1-0 [relax] 0-2-0-2-0-2-0-2-0-2-0-2-0-2-0-2-0 [relax] 0-3-0*… ,**sequential** - 6 action blocks, each block is a subsequent execution of all active gestures:*[relax] 0-1-0-2-0-3-0-6-0-7-0-8-0-9-0 [relax] 0-1-0-2-0-3-0-6-0-7-0-8-0-9-0 [relax] 0-1-0-2-0*… ,**repeats_short** - 7 action blocks, each block contains 6 repetitions of each active gesture:*[relax] 0-1-0-1-0-1-0-1-0-1-0-1-0 [relax] 0-2-0-2-0-2-0-2-0-2-0-2-0 [relax] 0-3-0*… .

This results in 20 repetitions of each active gesture in a single experiment performed for one participant. Each subject performed the experiment twice with at least one-week time separation, raising the number of active pose repetitions to 40. However, electrode band placement repeatability is not completely ensured and could differ slightly between experiments.

Suitable PC software was created in order to guide the participant through the procedure and record data. During the experiment, the subject was presented with a photo of the desired gesture to be performed (see Figure 3). The software also provided a preview of the next gesture and countdown to next gesture transition. Before each experiment, the subject performed a familiarisation procedure, which is not included in the dataset.

### 2.3. Participants

Each subject signed a participation consent form, along with a data publication permit. Before the first experiment, each participant filled a questionnaire that included gender, age, height, weight, dominant hand, health history, contact allergies, sports activity and tobacco usage. Additionally, forearm diameter was measured in places where wrist and elbow bands were placed. Anonymous questionnaires are published along with the putEMG dataset. The study was approved by the Bioethical Committee of Poznan University of Medical Science under no 398/17.

putEMG dataset covers 44 healthy, able-bodied subjects—8 females and 36 males, aged 19 to 37. As described in Section 2.2, each participant performed the experiment twice.

### 2.4. Pre-Processing and Labelling

Published sEMG data was not altered by any means of digital signal processing. Signals provided in the putEMG dataset are raw ADC values, not filtered or trimmed. Each trial is an uninterrupted signal stream, that includes all trajectory parts—steady states and transitions. If needed, the sEMG signal can be converted to millivolts using the following formula:(1)$$x=\frac{N\cdot 5}{2^{12}}\cdot \frac{1000}{200}\left[\mathrm{mV}\right],$$
where *N* is a raw ADC value stored in putEMG files.

The sEMG data are labelled by a trajectory that was presented to participants during experiments (**TRAJ_1** column in the dataset), with gestures labelled with numbers presented in Figure 3 (*relax* periods are marked as (-1)). However, as gesture execution may not perfectly align to given trajectories due to the subjects’ delay in command performance or mistakes, all gestures were relabelled based on the captured video stream. VGGNet [27], a convolutional neural network, was trained and applied to video gesture recognition with 98.7% accuracy (saved as **TRAJ_GT_NO_FILTER** column). Afterwards, the output from the neural network was filtered using a median filter and then referenced to given trajectory values. Each gesture execution was allowed to begin 250 ms earlier than the given user prompt and end 2000 ms later than the appearance of the next prompt. Any occurrences of a gesture outside of these boundaries were trimmed. Contradictory labels (where most of a particular gesture occurrence was recognised as mistaken) were labelled as (−1), marking them not to be processed. As participants could notice and try to correct the mistake mid-execution, which would cause abnormal muscle activity, the whole gesture occurrence is rejected. Filtered VGGNet gesture ground-truth were put into the **TRAJ_GT** column. Examples of ground-truth trajectories placed side by side with corresponding EMG recordings are presented in Figure 4.

putEMG dataset is available under Creative Commons Attribution-NonCommercial 4.0 International (CC BY-NC 4.0) license at: https://www.biolab.put.poznan.pl/putemg-dataset/. For more detailed information on data structures and handling see Appendix A.

## 3. Technical Validation

This section presents the validation of the putEMG dataset properties. Recorded sEMG signals were assessed in respect of their usefulness. Well established state-of-the-art classification methods and feature sets were examined as a baseline for future applications.

### 3.1. Amplitudes Assessment

Many factors affecting sEMG signal variability exist. Across subject population, main differences include skin-electrode contact impedance, the thickness of fat tissue and diversity in muscle activity level. A common approach in most EMG processing methods, to avoid the above issue, is signal normalisation with respect to Maximum Voluntary Contraction (MVC) [28] or another reference activity [4]. In this work, sEMG amplitudes are characterised as a ratio of signal power during active gestures with respect to *idle*, expressed as signal-to-noise ratio (SNR). This allows for direct comparison between signals in *idle* and during active gesture performance, which is crucial for classification purposes. The SNR is given by the formula:(2)$$SNR_{dB}=20\cdot log\left(\frac{P_{gesture}}{P_{idle}}\right),$$
where $P_{gesture}$ is the average signal power of active gesture, $P_{idle}$ refers to signal power in *idle* state. Respective signal sections were selected using trajectory after label processing and transition removal. SNR results per gesture averaged across 5 most active channels in each gesture are presented in Figure 5a. *Extension* gesture generates significantly (*p* < 0.05) higher activity than other hand movements, for the majority of subjects averaged SNR is calculated above 14 dB. SNR for *fist* and *flexion* is recorded above 11 dB, while for pinch gestures SNR is significantly lower, with median value falling in the range of 3 to 6 dB. Histograms of SNR for two substantially different gestures (*flexion* and *pinch middle*) are presented in Figure 5b,c, presenting quasi-normal distribution. However, distribution parameters differ significantly (*p* < 0.05). Smallest observed *flexion* gesture SNR, for most subjects, is above 10 dB. Only for 2 participants it has fallen below 10 dB. In the case of *pinch middle* gesture, for 3 subjects, SNR value was recorded below 0 dB. In the case of these participants, manual analysis of raw sEMG signals and video footage showed abnormal activity during *idle* stage, caused by excessive forcing of finger extension during these periods. In general, for all pinch gestures, lower SNR was observed. It is due to a smaller count of muscles being activated during these gestures, causing fewer channels to present high activity.

### 3.2. Feature Extraction and Classification Benchmark

The advantages of the putEMG dataset are a large number of subjects and a larger number of gesture repetitions compared to other publicly available datasets. The dataset can be used to develop classification methods or to extract or test subject-invariant features. Therefore, in this work, we present the performance of state-of-the-art classification methods and feature sets.

#### 3.2.1. Used Feature Sets and Classifiers

So far, many approaches have been presented to EMG feature extraction and classification [8,29]. For this work, three most popular feature sets were selected: Root Mean Square (RMS), Hudgins’ feature set [30] and Du’s feature set [31]. Hudgins’ set consists of Mean Absolute Value (MAV), Waveform Length (WL), Zero Crossing (ZC) and Slope Sign Change (SSC). It is characterised by relatively high classification accuracy, high insensitivity to window size and low computational complexity [32]. Du’s feature vector is composed of integrated Electromyogram (iEMG), variance (VAR), WL, ZC, SSC and Willison Amplitude (WAMP). In some research, it is reported to present higher performance than Hudgins’ set [29]. Even though Hudgins’ and Du’s sets are computed in the time domain, they represent amplitude, frequency, and complexity of the signal [8]. The gesture classification was performed using four different classifiers: Linear Discriminant Analysis (LDA), Quadratic Discriminant Analysis (QDA), k-nearest Neighbours Algorithm (kNN) and Support-Vector Machine (SVM) (with radial basis function (RBF) kernel), which are commonly used in EMG classification tasks [6,8,29,31]. Besides well-established, classic classification approaches, a number of attempts use DNNs [13]. In order to compare results generated by DNNs, all hyperparameters, network topology and training scheme has to be reproduced. Unfortunately, networks architecture and hyperparameters values cannot be simply deduced from most of the manuscripts. In this article, a comparative benchmark using DNN was not performed.

#### 3.2.2. Classification Pre-Processing and Data Split

Solely for classification purposes, raw sEMG data were filtered using a bandpass filter with cutoff frequencies of 20 and 700 Hz. Furthermore, to reduce interference generated by the mains grid and other equipment used during the experiment, an adaptive notch filter attenuating frequencies of 30, 50, 90, 60, and 150 Hz was applied. Filtered sEMG signal was used to calculated features needed for feature sets described in Section 3.2.1. A moving window of 5000 samples (488 ms) and step of 2500 samples (244 ms) was used during the above calculations. Gesture ground-truth label for calculated features was sampled at the end of the processing window. In addition to initial label pre-processing described in Section 2.4, the gesture trajectories were further trimmed to exclude transitions between *idle* and active gesture: 488 ms before the gesture start and 244 ms after the gesture start (see *After transition removal* plot in Figure 4). Trimming periods correspond to analysis window length and step used in feature calculation, as each feature sample is computed using past raw data.

Each participant’s data, from each experiment day, were processed separately generating 88 independent subsets. Among each subset, three trials were present, as described in Section 2.2. These trials were divided into training and test datasets, generating 3-fold validation splits, where two trials were concatenated to generate a training set and the remaining trial was used as a testing set.

#### 3.2.3. Classification Performance

This section presents a comparison between different combinations of classifiers and feature sets, all differences described here are significant with a probability value of *p* < 0.05. Significance of classifier mean accuracy comparisons was evaluated using ANOVA and Tukey’s tests. Precision, recall, and accuracy were calculated for each class separately and averaged with equal weights.

Used classifiers’ hyperparameters were empirically tuned in order to minimise the difference in accuracy between training and test sets. For the SVM classifier, *C* parameter of the RBF kernel can be treated as regularisation parameter tuning decision function’s margin. Large values of *C* will reduce acceptance margin, while lower values will promote wider margin and in consequence, generate a simpler decision function. In this work, C=50 was selected. kNN *k* parameter reflecting the number of neighbouring samples was set to 5. For QDA, classifier covariance estimate regularisation parameter was equal to 0.3. In case of default parameters for SVM (C=1.0), underfitting was observed, while for QDA (regularisation parameter equal to 0.0), the results were overfitted.

Among tested classifier and feature sets, the highest overall mean accuracy was achieved by LDA/Hudgins, LDA/Du, and SVM/RMS combinations (89–90%, see Figure 6), differences between them are insignificant. For RMS feature, the accuracy of SVM is better (90%) than for LDA, kNN, and QDA (86%, 87%, 81% respectively). Difference between kNN and LDA is negligible (Figure 6). Accuracies achieved for Hudgins’ and Du’s feature sets within a single classifier are very similar; however, results between various classifiers fed with Hudgins’ or Du’s sets differ significantly. For QDA, kNN, and SVM classifiers, combined with the RMS feature, accuracy presents exceeding performance (81%, 87%, 90% respectively) compared with other features sets (74%, 78%, 87%). Unlike other classifiers, LDA presents higher accuracy for Du’s and Hudgins’ feature sets (88% and 89%) than for RMS (Figure 6).

Figure 7 presents the resulting precision-recall relationship of tested classifiers and feature sets combinations. LDA/Du and LDA/Hudgins combinations display similar performance, significantly different than all remaining classifiers. Highest precision score (89%) is presented by SVM/RMS. Additionally, the precision scores of LDA/Du and LDA/Hudgins are insignificantly different with respect to SVM/Du, SVM/Hudgins and LDA/RMS. The highest recall was achieved for SVM/RMS, LDA/Du and LDA/Hudgins (83%, 84%, 82%).

Figure 8 presents precision score with respect to each gesture and electrode setups consisting of 24 and 8 sEMG channels. Results are presented only for three classifier and feature set combinations showing the highest accuracy (see Figure 6). Considering 24 electrode configuration (Figure 8a), LDA/Du composition has higher precision (92%) for *idle* than other considered classifiers (90–91%). LDA/Du and LDA/Hudgins present similar precision for *fist*, *flexion* and *extension* gestures (93%, 95%, 95%) than SVM/RMS (91%, 93% and 93%). Conversely, SVM/RMS has a higher precision score for *pinch index*, *pinch middle* and *pinch ring* gestures (75%, 85%, 90%) than LDA/Du or LDA/Hudgins (68%, 79%, 87%). The differences in precision score classification for *pinch small* gesture are insignificant (89–86%). For all classifiers, the lowest precision is achieved for *pinch index* gesture. In case of 8 sEMG channel configuration (middle electrode band, Figure 8), *idle* state is detected with lower precision than for 24 channels setup, LDA/Du combination remains better performing (89%) compared to all other classifier and feature sets. For remaining gestures, differences between LDA/Du and LDA/Hudgins are insignificant. These classifiers perform with higher precision while detecting *fist*, *flexion* and *extension* gestures (90%, 94%, 96%) compared to combinations including SVM (86%, 90%, 92%). For pinch gestures, the highest precision is presented by SVM/RMS (52%, 68%, 79%, 79%); however, compared to LDA/Du, the difference is insignificant (49%, 62%, 77%, 74%). Precision for LDA/Hudgins, in the case of pinch gestures, is lower than for SVM/RMS.

The decay in precision between 24 and 8 channel configurations for *fist*, *flexion* and *extension* gestures is insignificant when considering LDA classifiers, although it is notable for SVM/RMS combination in 24 electrode setup. Still, an absolute decrease in precision is smaller than 5%. For all classifiers, decay in precision for pinch gestures classification can be observed (10–23%).

## 4. Discussion

Mean signal powers averaged across all records reveal that *fist*, *flexion* and *extension* generate stronger activity than pinch gestures, by more than 4 dB. Significant differences in amplitudes between tested gestures were also reported by Phinyomark et al. [33] but in contrast to our results, the *fist* gesture generated significantly stronger activity than *flexion* or *extension*. Contrary to all the above, the results from NinaPro dataset did not reveal significant differences in amplitudes with respect to gestures [15]. The origin of these discrepancies is ambiguous. In the case of NinaPro dataset, a higher activity might be induced by the stiffness of CyberGlove II, whereas in the case of the putEMG dataset, hand motion was not obstructed. Another possible explanation is differences in the experimental procedure and data processing. In contrast to NinaPro, in our experiment, the elbow was always supported. Moreover, our results are presented as SNR with respect to *idle* state, and therefore activity generated by the gravitational load is filtered out, while in NinaPro observed activity contains both the component caused by volatile movement and hand weight compensation.

A significant difference between pinch gestures and other gestures is observed in both signal amplitudes and classification results. Individual finger motion generates significantly smaller muscle engagement than full hand motions, EMG activity area is also limited [34]. Therefore, for distinguishing between pinch gestures, methods able to map arbitrary regions of activity may be more feasible. It might explain why SVM/RMS performs better for pinch gestures than LDA, even when the latter uses more complex feature sets. SVM is able to generate more complex decision boundary to distinguish between different spatial activity patterns, compared to LDA. In this case, richer information related to muscle activity, presented by Du’s or Hudgins’ feature sets, is less beneficial than complex spatial decision boundary. This statement can also be proven by significant degradation of SVM/RMS precision when only 8 electrodes are considered.

Based on the ROC curve for a given classifier, precision and recall are dependent in the sense that increasing one of these measures decreases the other one. Even though Figure 7 does not present ROC curves but only single points referring to classifier/feature set configurations, it can be deduced that SVM/RMS, LDA-Hudgins, and LDA/Du setups perform better than other classifiers. Based on overall accuracy, none of them can be pointed out as superior at a significant level (*p* < 0.05). Precision scores calculated for each gesture separately reveal that LDA/Du is superior for classification of *idle*, *fist*, *flexion*, and *extension*, while SVM/RMS performs better for pinch gestures.

Reported putEMG gesture classification performance is lower than results presented in similar works, where sEMG signals were recorded for smaller groups of subjects [6,8,35,36,37]. Kushaba et al. [6] analysed only full hand movements and *pinch index* gesture and reported highest accuracy (98–99%) for SVM classifier, depending on the subject’s pose, and 97–99% with LDA. Al-Timemy et al. [35] analysed individual finger motions and reported classification accuracy of over 98% for 5 classes and over 90% for 12 classes. With putEMG overall accuracy of 90% was achieved. The discrepancies can be explained in several ways. In the aforementioned works, different feature vectors were used and the Orthogonal Fuzzy Neighbourhood Discriminant Analysis (OFNDA) dimensionality reduction method was applied. In our results, the dimensionality reduction issue is omitted, as the classification is performed as technical validation of putEMG dataset. In works mentioned above, dimensionality reduction was performed for each training set separately, which could lead to drawing improper conclusions regarding full dataset. The second crucial difference is the number of subjects and the experimental procedure. In the case of References [6,8,35,36,37], no more than 10 subjects participated in the experiment. In broader datasets, there is a higher chance that subjects participated in such study for the first time, therefore the diversity of the group may be higher and degrade overall accuracy. As can be seen in Figure 8a,b, median precision values are higher than reported mean values, with mean values often closer to the 25th percentile, suggesting the drop in average performance caused by individual subjects. Gesture diversity also has a high impact on obtained results. In presented dataset validation, both individual finger gestures and full hand movements are recognised by a common classifier, while in mentioned experiments, they were processed separately. A higher number of gesture classes may cause accuracy decay. For instance, in the case of NinaPro DB5 dataset and 41 gesture classes, reported accuracy was 74% [17].

The RMS feature performs better with QDA, kNN, and SVM classifiers due to the low size of input vector (24×1), while Hudgins and Du sets consist of 24×4 and 24×6 features, respectively. Tested classifiers differ in the complexity of the decision boundary, and in the number of parameters to be estimated during the training process. In the case of LDA, the number of parameters is the lowest and proportional to the number of inputs. For QDA covariance is estimated separately for each class and the number of parameters is quadratic in relation to input vector size. In the case of kNN, the number of parameters affects metric confidence, as features are not weighted with respect to their decisive importance. It indicates the necessity of dimensionality reduction of the input feature set before the classification.

After tuning, hyperparameters values revealed that regularisation is required to avoid overfitting. Nevertheless, the selected number of kNN neighbours is 5, a low value considering 8 gestures and at least 10,000 samples per gesture in the training set. In the case of SVM, the *C* parameter had to be set to a high value promoting small margins and more complex decision boundary. These facts might indicate that the dataset is highly diverse. It might be thanks to the procedure consisting of gestures performed both in sequences and consecutive repeats, and variable gesture durations.

## 5. Conclusions

This paper presents an sEMG signal dataset intended for hand gesture recognition purposes, acquired for 44 able-bodied subjects performing 8 different gestures (including *idle* state). A publicly available putEMG dataset is aimed to be a benchmark for classification approach comparison and as a test bench for evaluation of different aspects of wearable sEMG-based HMIs, such as electrode configurations, robustness improvement or armband localisation changes. The most significant advantage of the presented putEMG dataset is a large group of participants performing a high number of gesture repetitions compared to similar work [8,15,19]. High gesture performance diversity was assured by including both repetitive and sequential gesture execution procedure with varying duration. The dataset also introduces a novel approach to gesture execution ground-truth by utilising an RGB video stream and a depth camera.

The dataset validation, using state-of-the-art classification methods, demonstrated an acceptable classification accuracy of approximately 90% (for SVM/RMS, LDA/Hudgins and LDA/Du classifier and feature set combination). Although this result is worse than those presented in related work, it was achieved for a larger and highly diverse group of participants. Moreover, dimensionality reduction was not applied, which might also affect classification performance. It should be underlined that the main aim of this article was to present the dataset, and classification results have to be treated as a baseline for characterising putEMG data quality and not classifier performance itself. The obtained results indicate that putEMG can be successfully used as a benchmark for wearable, end-user hand gesture recognition systems development.

## Figures and Tables

**Figure 1 sensors-19-03548-f001:**
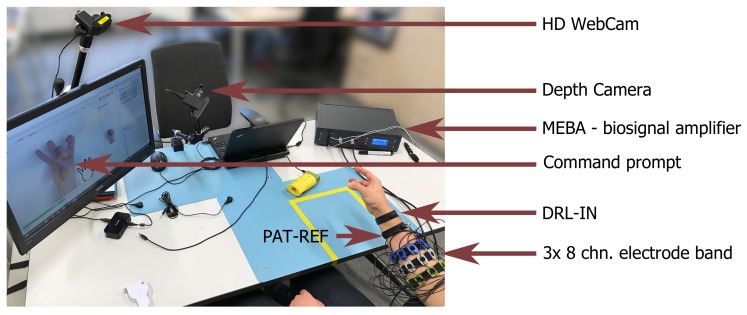
Setup used during the acquisition of putEMG dataset; visible placement of sEMG sensor bands and reference electrodes located near the subject’s wrist.

**Figure 2 sensors-19-03548-f002:**
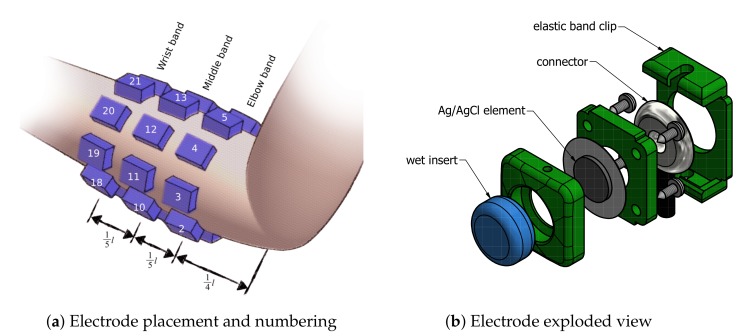
Schematics of electrode placement on the subject’s forearm during the experiment, and design of the reusable wet electrode setup used for the putEMG dataset experiment.

**Figure 3 sensors-19-03548-f003:**
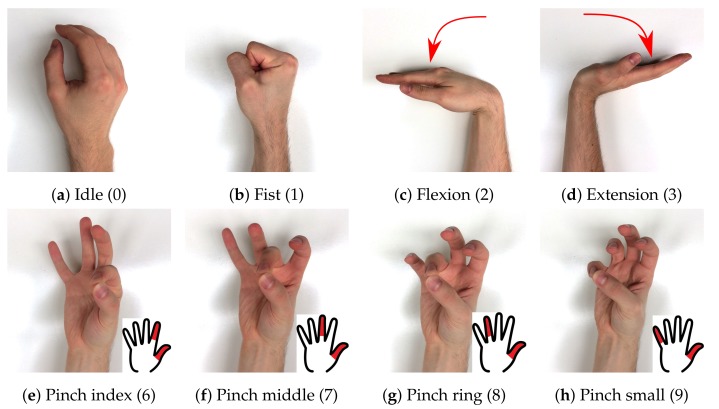
putEMG includes 8 gestures; 7 active gesture and *idle*, where subjects were asked not to move their hand; numbers in square brackets indicate gesture marker used in putEMG dataset files.

**Figure 4 sensors-19-03548-f004:**
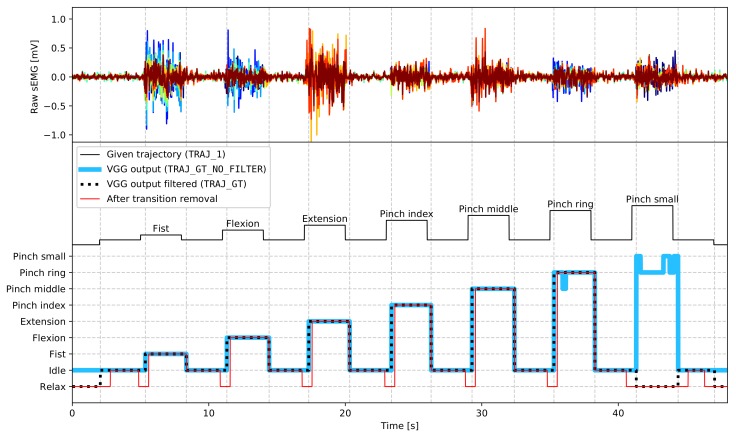
Exemplary fragment of ground-truth label processing stages; *fist*—*pinch middle*: gestures were performed and recognised properly, only transition points were adjusted; *pinch ring*: the gesture was mostly recognised as correct, mistakes in *VGG output* were filtered; *pinch small*: the gesture was mostly recognised as incorrect, the whole gesture was rejected; vertical grid corresponds to transitions in *VGG output filtered.*

**Figure 5 sensors-19-03548-f005:**
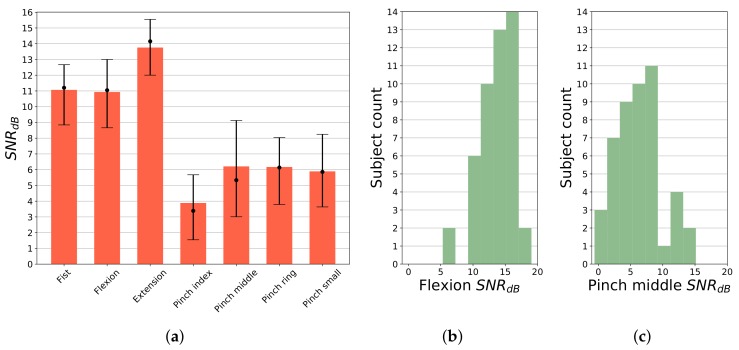
SNR analysis of gesture execution; all values were calculated relative to *idle* period, in this case considered as noise; for each gesture type SNR was averaged over 5 most active channels; (**a**) SNR by gesture; bars represent mean value; dot marks SNR median value together with 25th and 75th percentile; (**b**) Histogram of SNR for *flexion* gesture; (**c**) Histogram of SNR for *pinch middle* gesture.

**Figure 6 sensors-19-03548-f006:**
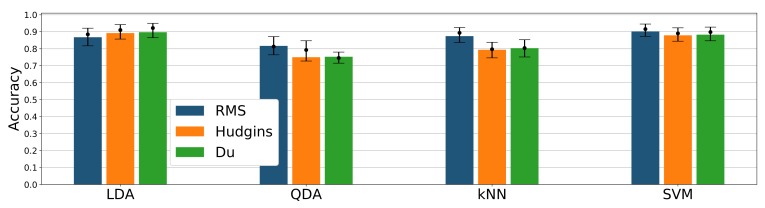
Classification accuracy for different classifiers and feature sets combinations; bars represent mean value; dots mark accuracy median value together with 25th and 75th percentile; results presented for 24 channel configuration.

**Figure 7 sensors-19-03548-f007:**
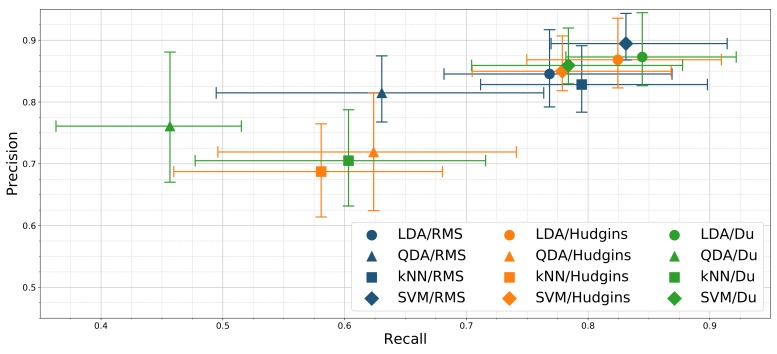
Classification precision and recall mean relationship for different classifiers and feature sets combinations; error bars mark 25th and 75th percentile; results presented for 24 channel configuration

**Figure 8 sensors-19-03548-f008:**
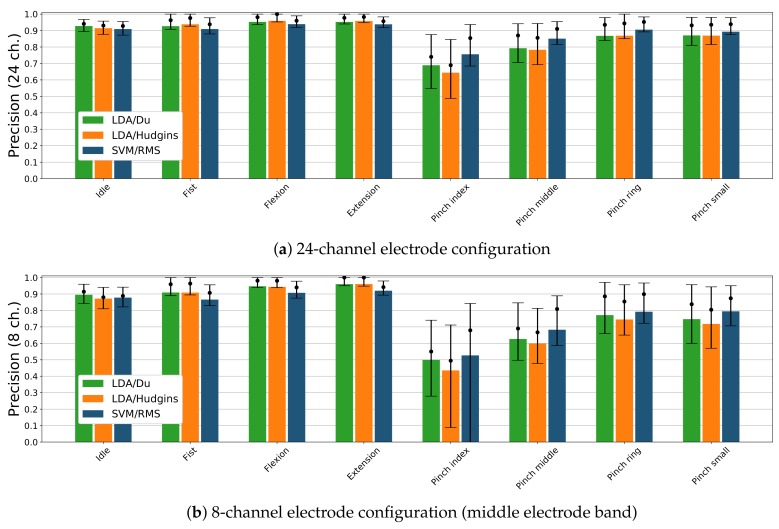
Classification precision for selected classifiers and feature sets combinations, presented for each active gesture in the putEMG dataset; bars represent mean value; dots mark precision median value together with 25th and 75th percentile.

**Table 1 sensors-19-03548-t001:** A summary of publicly available sEMG datasets of hand gestures; the list only contains records for able-bodied subjects; biosignal recording systems: ${}^{a}$ OttoBock, ${}^{b}$ Delsys, ${}^{c}$ Cometa, ${}^{d}$ Thalmic Labs Myo, ^e^ Delsys Trigno, ${}^{f}$ EMG System do Brasil, ${}^{g}$ OT Bioelettronica MEBA, ${}^{h}$ Delsys Bagnoli; gesture tracking systems: ${}^{1}$ CyberGlove II, ${}^{2}$ Tobii Pro Glasses 2, ${}^{3}$ RGB HD Camera, ${}^{4}$ Depth sensor–Real Sense SR300; ${}^{†}$ including different objects, ${}^{‡}$ including 2–4 different objects.

Dataset Name	EMG Recording Setup	Gesture Tracking System	No. of Participants	No. of Gestures	Repetitions per Session	Session Count	Session Organisation and Intervals	Trials Organisation	Gesture Durations
NinaPro DB1 [15]	10 sEMG ${}^{a}$	yes ${}^{1}$	27	52	10	1	-	random	gesture: 5 s idle: 3 s
NinaPro DB2 [16]	12 sEMG ${}^{b}$	yes ${}^{1}$	40	49	6	1	-	repetitive	gesture: 5 s idle: 3 s
NinaPro DB4 [17]	12 sEMG ${}^{c}$	-	10	52	6	1	-	repetitive	gesture: 5 s idle: 3 s
NinaPro DB5 [17]	16 sEMG ${}^{d}$	yes ${}^{1}$	10	52	6	1	-	repetitive	gesture: 5 s idle: 3 s
NinaPro DB6 [10]	14 sEMG ^*e*^	yes ${}^{2}$	10	7	12 ${}^{†}$	10	2 per day, 5 days	repetitive	gesture: 4 s idle: 4 s
NinaPro DB7 [22]	12 sEMG ^*e*^, 9DoF IMU	-	20	40	6	1	-	sequential	gesture: 5 s idle: 5 s
IEE EMG [19]	12 sEMG ${}^{f}$	-	4	17	32	1	-	sequential (4 varying)	no idle phase
Megane Pro [20]	14 sEMG ^e^	yes ${}^{1,2}$	10	15	12 ${}^{‡}$	10	2 per day, 5 days	repetitive	gesture: 8 s idle: 4 s
EMG Dataset 2 [23]	8 sEMG ${}^{h}$	-	8	15	12	3	-	sequential	gesture: 20 s
EMG Dataset 6 [6]	7 sEMG ${}^{h}$	-	11	8	12	6	5 poses	sequential	gesture: 5 s idle: 3-5s
CapgMyo(DB-a) [21]	128 HD-sEMG	-	18	8	10	1	-	repetitive	gesture: 3–10 s idle: 7 s
CapgMyo(DB-b) [21]	128 HD-sEMG	-	10	8	10	2	1 day	sequential	gesture: 3 s idle: 7 s
CapgMyo(DB-c) [21]	128 HD-sEMG	-	10	12	10	1	-	repetitive	gesture: 3 s idle: 7 s
CSL-HDEMG [24]	192 HD-sEMG	yes ${}^{1}$	5	27	10	5	different days	sequential	gesture: 3 s idle: 3 s
**putEMG** **(this work)**	**24 sEMG** ${}^{g}$	**yes** ${}^{3,4}$	**44**	**8**	**20**	**2**	**1 week**	**sequential** **repetitive**	**gesture: 1 s or 3 s** **idle: 3 s**

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
