# Peer review of "putEMG—A Surface Electromyography Hand Gesture Recognition Dataset"

_sensors, 2019, doi:10.3390/s19163548_

Round 1

Reviewer 1 Report

This work describes a public database of EMG signals for active hand movements and presents a pattern recognition system for its validation. Datasets were captured from 44 able-bodied participants, 24 sEMG channels with the tracking system, 8 gestures in three trajectory configurations, two sessions, for a total of 40 repetitions. Gestures were held for different times and different sequences of the user prompt were considered. The database is very well described, it includes relevant aspects about the movements for label definition and important considerations were reported about the final ground-truth trajectory. A validation of gestures dataset has been proposed to demonstrate the proximity with the expected in the literature. A pattern recognition system with selected state-of-the-art feature-sets and classifiers was applied. Results compared overall performance for three different classifiers combined with three feature-sets. Moreover, individual gesture performance was also compared for selected combinations of classifier-features.

The main contribution of this work seems to be the database, which overcomes some of the lacks of the publicly available sEMG datasets described in the manuscript. Moreover, a novel approach to ground-truth of gesture execution is presented.

I have a few comments to help the authors improve the quality of the manuscript.

In the methodology description, the sampling of HD Web Camera and technical specifications of deep images are missing. Also, the methodology may include details about the detection of the subject's mistakes combining both video feed and deep images. Moreover, the sampling frequency defined for sEMG data is higher than a common 1kHz for many works in the literature. However, a higher frequency may imply a higher difficulty to the approximation to the real instant for onset and offset detection. Is the sampling frequency of the tracking trajectory higher than the dataset to be labeled, and how the authors can explain this choice? According to the text (line 174), the trajectory after transition removal was obtained from outputs of the DNN applied for a video gesture recognition. Which is the relation of the outputs of the DNN and the sEMG samples? Is there an expected error of identification of onset and offset of the gestures?

In lines 176-177, (Gesture execution detection contained in between boundaries of given user prompt was accepted to the final ground-truth trajectory). How much time was considered for boundaries between user prompts?

Description about ground-truth trajectory is not very clear. 

How can be explained the unbalance number of participants related to females/males?

Ages should include the average value and standard deviation. The same aspect should be included for all the values presented in the Technical validation session, for accuracy, precision and recall values.

The label of x-axis in Fig 4 is missing.

In line 179, it was stated that contradictory labels were gestures recognized as mistaken. This situation was related to the accuracy of the DNN? Manual detection using frames from HD web camera can be used instead the video recognition to improve the accuracy of labels? Authors could explain the advantage of the use of DNN and to show the comparison with traditional techniques for onset/offset detection.

Line 190. Pgesture was calculated using boundaries?

Line 262. Check the accuracies order for the classifiers, respectively.

Lines 272 and 279. Case of 8 sEMG channels is only explained in the legend of Figure 8b but no inside the text.

Line 329. I suggest defining OFNDA in the text (Orthogonal Fuzzy Neighbourhood Discriminant Analysis).

Author Response

In the methodology description, the sampling of HD Web Camera and technical specifications of deep images are missing. Also, the methodology may include details about the detection of the subject's mistakes combining both video feed and deep images. Moreover, the sampling frequency defined for sEMG data is higher than a common 1kHz for many works in the literature. However, a higher frequency may imply a higher difficulty to the approximation to the real instant for onset and offset detection. Is the sampling frequency of the tracking trajectory higher than the dataset to be labeled, and how the authors can explain this choice? According to the text (line 174), the trajectory after transition removal was obtained from outputs of the DNN applied for a video gesture recognition. Which is the relation of the outputs of the DNN and the sEMG samples? Is there an expected error of identification of onset and offset of the gestures?

Response: We have decided to present video and depth stream details (resolution, formats etc.) in the Appendix entitled “putEMG dataset structure and handling” as they are not the main scope of the article. Additionally, after revision frame-rate of both video and depth stream was added.

The sampling frequency is caused by predefined settings available in used amplifier/data acquisition device. 5120 Hz was the lowest value guaranteed to avoid aliasing. No downsampling or any other EMG signal preprocessing was used to provide database users with raw, unmodified data.

Sampling rate of gesture tracking is equal to the framerate of chosen RGB camera (30 fps), which is assumed to be sufficient considering movement speeds. The DNN was used mainly as a means of detecting participants’ mistakes. Of course, there will always be a delay between muscle activity and mechanical hand movement (hence, information visible in the video feed). Moreover, the exact moments of gesture beginnings/ends will always be a subject to arbitrary thresholding, regardless of used method. If more exact gesture instants are required, for example to analyse the transitions, they can be extracted from the EMG signal itself.

In lines 176-177, (Gesture execution detection contained in between boundaries of given user prompt was accepted to the final ground-truth trajectory). How much time was considered for boundaries between user prompts?

Response: The allowed boundary started 250 ms before particular gesture prompt and ended 2000 ms after the following “idle” prompt was shown. The description has been rephrased for clarity (around line 180).

Description about ground-truth trajectory is not very clear. 

Response: The description was rephrased in several places, hopefully that makes it more clear.

How can be explained the unbalance number of participants related to females/males?

Response: The volunteers were picked at random, however, the environment they came from could be biased towards males.

Ages should include the average value and standard deviation. The same aspect should be included for all the values presented in the Technical validation session, for accuracy, precision and recall values.

Response: Detailed list of participants ages is provided alongside the database.

Accuracy/precision/recall values here cannot be properly described with normal distribution parameters (e.g. because of hard upper limit of 100%). Mean values presented in text, along with median and 25th/75th percentiles in the charts, were found to be much more descriptive while not cluttering the text.

The label of x-axis in Fig 4 is missing.

Response: Added the label.

In line 179, it was stated that contradictory labels were gestures recognized as mistaken. This situation was related to the accuracy of the DNN? Manual detection using frames from HD web camera can be used instead the video recognition to improve the accuracy of labels? Authors could explain the advantage of the use of DNN and to show the comparison with traditional techniques for onset/offset detection.

Response: The DNN was trained in several iterations. The training set was carefully chosen and manually relabeled to achieve the highest performance. The final iteration of DNN was similar in recognition accuracy to manual labeling, in some cases, not even a human can perfectly determine between different pinch gestures, as fingers cover each other. Post-process verification of DNN performance has revealed that in majority of cases, labels marked as contradictory (when DNN label did not match the given experiment trajectory), pointed to a time in recording where the subject made a mistake, confused the gesture or changed the active gesture during the given action phase. Appropriate comment/description concerning contradictory labels was added in label processing description. Details of DNN training, verification etc. were omitted in text as this is not the main scope of the article.

There is no clear advantage in using DNN for gesture transition detection in comparison with traditional methods. However, the DNN approach provides additional information on subjects’ mistakes. In an experiment involving a group as large as in case of putEMG, it is impossible for the supervisor to control all participant’s actions. One badly executed gesture can significantly impair the classifier training performance, hence blur the results.

Line 190. Pgesture was calculated using boundaries?

Response: Yes, appropriate information was added in SNR formula description.

Line 262. Check the accuracies order for the classifiers, respectively.

Response: Excellent find, fixed listing order.

Lines 272 and 279. Case of 8 sEMG channels is only explained in the legend of Figure 8b but no inside the text.

Response: Added missing reference to Figure 8b, in line 282. Added “middle electrode band” description in text.

Line 329. I suggest defining OFNDA in the text (Orthogonal Fuzzy Neighbourhood Discriminant Analysis).

Response: Added OFNDA to glossary, added definition in text.

Reviewer 2 Report

no further comment, it is a very interesting article, and well-prepared either on materials and on further EMG signal processing and classification. 

Author Response

no further comment, it is a very interesting article, and well-prepared either on materials and on further EMG signal processing and classification.

Thank you.

Reviewer 3 Report

This paper presents a new EMG dataset for benchmarking hand gesture recognition methods based on sEMG signal. The dataset provides sEMG data from 44 participants which is a large increment from most existing datasets. The reviewer feels that this dataset will benefit many researchers that need to compare the effectiveness of their classification algorithms. The manuscript is generally well written but some technical details should be added to improve the reproducibility and readability. The reviewer suggest the authors address the following comments.

The reviewer suggests the authors revise the term 'subjects' to 'participants'.

Although the dataset provides sEMG data from many participants, the number of type of gestures are relatively small. Are there other advantages of the dataset apart of the high number of participants? (How would the author persuade the researchers to use your dataset?)

The gesture address in this manuscript includes fist, flexion, extension, pinching of index, middle ring and small fingers, and idle state. Why are these gestures selected? From the reviewers point of view, since the number of gestures are relatively small, shouldn't the authors have chosen gestures that frequently occur in daily activities?

An illustration depicting the sequence of how the gestures are performed would be helpful. By the way, are the sequence of the gesture random?

Were there any rest sessions between each repetitions?

Were there any criteria in the selection of participants?

How is the accuracy calculated? (Is is calculated with respect to time?) Please clarify.

Author Response

The reviewer suggests the authors revise the term 'subjects' to 'participants'.

Response: The words were used interchangeably to avoid multiple repetitions.

Although the dataset provides sEMG data from many participants, the number of type of gestures are relatively small. Are there other advantages of the dataset apart of the high number of participants? (How would the author persuade the researchers to use your dataset?)

Response: Apart from large number of participants:

- a large number of repetitions of each gesture can be highlighted,  40 repetitions for each gesture is among top of listed databases
- the dataset includes both repetitive and sequential trajectories, which helps to prevent overfitting
- each participant performed twice, with one-week interval, with electrode bands worn in a similar way, but not positioned exactly the same

These qualities facilitate development of everyday-use EMG HMIs, where the classifier must be generalized for a larger population, the user wears the device slightly different each time, and the characteristics of signal change over time.

The gesture address in this manuscript includes fist, flexion, extension, pinching of index, middle ring and small fingers, and idle state. Why are these gestures selected? From the reviewers point of view, since the number of gestures are relatively small, shouldn't the authors have chosen gestures that frequently occur in daily activities?

Response: The particular gestures were chosen based on state-of-the-art research and authors’ previous experience, to be easily recognizable and distinguishable from each other. From an end-user HMI perspective, the gestures should actually highly differ from daily activities, to prevent accidental command triggering.

An illustration depicting the sequence of how the gestures are performed would be helpful. By the way, are the sequence of the gesture random?

Response: Added visual (text) representation of all trajectories in putEMG dataset (repeats_long, sequential and repeats_shorts) in section with trajectory description (2.2 Procedure). Additionally, example of sequential pattern can be seen in Figure 4. Gestures in trajectory are not randomised.

Were there any rest sessions between each repetitions?

Response: Yes, there were 10-second relax periods between action blocks.

Were there any criteria in the selection of participants?

Response: The volunteers were picked at random, the only requirement was to have no major physical disorders related to right arm.

How is the accuracy calculated? (Is is calculated with respect to time?) Please clarify.

Response: Added “Precision, recall, and accuracy were calculated for each class separately and averaged with equal weights.” for clarification in section 3.2.3 Classification performance.

Round 2

Reviewer 1 Report

The authors have addressed all my comments and suggestions.

Minor Concerns

148. Reword 'an subsequent' to 'a subsequent'.

Reviewer 3 Report

The reviewer would like to thank the authors for addressing the comments. All comments have been dealt with appropriately and the manuscript has been significantly improved.